# The impact of the digital economy on industrial structure upgrading in resource-based cities: Evidence from China

Zhenqiang Li[1‡], Qiuyang Zhou[2‡]*, Ke Wang[3]

1 School of Economics and Management, Hunan Institute of Science and Technology, Yueyang, Hunan, China, 2 School of Urban and Regional Science, Shanghai University of Finance and Economics, Shanghai, China, 3 School of Economics, Zhejiang Gongshang University, Hangzhou, Zhejiang, China

‡ ZL and QZ contributed to the work equally and should be regarded as co-first authors.
* zqy8862@126.com

**Data Availability Statement:** All relevant data are within the paper and its Supporting Information files.

**Funding:** Role of Funder statement: Zhenqiang Li: Data, Conceptualization, Funding Acquisition

## Abstract

The digital economy provides a new path to promote industrial structure upgrading. Using panel data from 2011 to 2020 for 85 resource-based cities in China, this paper empirically investigates the impact of the digital economy on industrial structure upgrading and the primary mechanism. The results show that the digital economy is conducive to promoting industrial structure upgrading in resource-based cities, and innovation is the primary mechanism of action. According to the different stages of resource development, we classify resource-based cities into growth, maturity, decline, and regeneration cities, and we further analyze the heterogeneous influence. In terms of influence degree, the digital economy has a more prominent role in promoting industrial structure upgrading in resource-exhausted cities. In addition, we also found that the closer to the provincial capital city, the more pronounced the promotion of the digital economy to the industrial structure upgrading.

## 1 Introduction

Since the Industrial Revolution, society's need for resources like coal and oil has grown, leading to the emergence of resource-based cities. These cities have grown by utilizing and processing the area's natural resources [1, 2]. Its firm reliance on resources is one of its distinguishing characteristics [3]. Most resources, such as coal and iron ore, are non-renewable and will soon run out [4]. The resource reserves of many resource-based cities have gradually decreased over the past few decades because of the ongoing exploitation of natural resources. Resource-based industries, an essential part of the economic systems of resource-based cities, are also declining because of the steady depletion of resources [5, 6]. Population loss, unemployment, environmental pollution, homogeneous industrial structure, and stagnant economic growth are a few issues that are becoming apparent [7–9]. The long-term dominance of resource-based sectors makes it harder to develop other industries [10]. Therefore, the growth of resource-based cities has more difficult obstacles [11], and it has drawn attention worldwide.

Investigation Qiuyang Zhou: Software Writing, Original Draft Writing, Review & Editing, Formal Analysis Ke Wang: Methodology, Writing, Review & Editing, Formal Analysis Financial Disclosure This work was supported by Key Scientific Research Project of Hunan Provincial Department of Education; The Grand/Award Number is 22A0468.

**Competing interests:** The authors have declared that no competing interests exist.

Based on the development experience of some countries, for resource-based cities to transform, the industrial structure is crucial. It is undeniable that natural resources have played a driving role in the economic growth of resource-based cities. However, the resource curse effect has become increasingly apparent in recent years; while natural resources constitute the comparative advantages of regions, they also create imbalances in regional industrial structure. In resource-based cities, long-term resource development has formed an industrial structure dominated by natural resources development and primary processing. As a result, the industrial structure of resource-based cities is relatively homogeneous and elementary. This industrial structure dominated by resource industries formed under the advantage of resource endowment increases the difficulty for cities to realize sustainable development. Thus, industrial structure upgrading is of great significance in the sustainable development process of resource cities. Specifically, in terms of economic development, in the process of industrial structure upgrading, the rapid emergence of new high-tech and service industries can inject new vitality into the economic development of resource-based cities. In terms of employment, upgrading the industrial structure can bring more employment opportunities, especially the development of new industries, which can absorb more labor and promote employment. In addition, in the context of resource scarcity, industrial structure upgrading is also conducive to enhancing the utilization efficiency of resources and promoting the optimal allocation of resources. Overall, industrial structure upgrading is closely related to the long-term development of resource cities. It is necessary to explore the practical path of upgrading the industrial structure of resource cities.

To encourage sustainable growth in resource-based cities, optimizing and regulating the industrial system is essential [12]. The focus of current discussions on the development of resource-based cities includes energy and environmental studies [13–16], evaluation of transformation development [17], the summary analysis of transformation policies and implementation effects [18]. The existing literature has laid the foundation for subsequent research on resource-based cities, but the analysis of this specific component of industrial structure upgrading needs to be further expanded.

The advancement of digital technology, the center of the latest technological revolution, is gaining momentum. The global economic environment is now significantly influenced by the digital economy, which unquestionably opens up new development potential for cities dependent on natural resources. Sustainable development requires information and digitization [19]. What impact has the digital economy had on the industrial structure of resource-based cities? What is the heterogeneity of this impact? What is the primary action path?

The modernization of resource-based cities' industrial structures has an apparent impact on the development of the national economy. In China, there are many resource-based cities. Therefore, we examine the effects of the digital economy on the modernization of the industrial structure, selecting 85 typical resource-based cities in China as the research object. The pertinent research findings help encourage to upgrade industrial structures in resource-based cities. This paper makes three contributions compared to earlier studies: (1) Most current research on resource-based cities focuses on ecological environment study and development appraisal [20, 21]. Research on other content needs to be expanded. We study the effects of the digital economy on industrial structure upgrading. The related conclusions can provide a new reference path for the industrial structure upgrading of resource-based cities. (2) From the perspective of technological innovation, we also examine the primary mechanism of action. (3) Depending on the development stage and the distance from the provincial capital, we carried out a heterogeneity analysis to recognize the impact of the digital economy on industrial structure upgrading in resource-based cities entirely, enriching the current study.

## 2 Literature review

### 2.1 Transformation of resource-based cities

Many scholars have studied the transformation paths of resource-based cities [22–24]. Ecological science, innovation, urbanization development, technology, and political incentives all have some influence on the transformation of resource-based cities. The government plays a crucial role in the transformation development process of resource-based cities [25, 26]. On the one hand, an appropriate compensation mechanism is essential for resource-based cities [27]. Environmental regulation, policy compensation, and financial help are essential for transforming resource-based cities [28]. In China, government transfer payments promote low-carbon development in resource-based cities [29]. On the other hand, laws and regulations can improve the transition performance of resource-based cities by constraining people's behavior [6], and the effectiveness of laws and regulations largely depends on the government. In addition, national policies often have a crucial influence on urban transformation [30]. Several scholars have studied the effects of some policies implemented in China. For example, the civilized city policy has significantly improved energy efficiency in resource-based cities [31]; the resource-based sustainable development policy has contributed to the economic, social, and ecological transformation of resource-based cities [32]; the new energy demonstration city policy has a significant effect on the green total factor productivity of resource-based cities [33].

### 2.2 The digital economy and industrial structural upgrading

Industrial structure upgrading implies the process of evolution from primary to advanced level [34]. With the acceleration of the global digitalization process, the digital economy has become an essential engine for promoting regional industrial structure upgrading [35], and resource-based cities are no exception. Due to the excessive reliance on natural resources, the industrial structure of most resource-based cities is relatively single, which crowds out the space for the development of new industries to a large extent. Moreover, the industrial chain of resource-based cities is not long, and there is extreme lack of high-value-added products, which also restricts the space for upgrading the original industries. The digital economy has injected a new driving force into the industrial structure upgrading of resource-based cities. On the one hand, digital technology is the core driving force of the digital economy [36]. The application of technologies such as big data and artificial intelligence has realized the intelligence of the production process of traditional industries, promoted the digital transformation of traditional industries [37], and driven the upgrading of the value chain [38]. Second, digital finance improves financial services coverage and broadens enterprises' financing channels in resource-based cities [39]. The broadened coverage and increased financing channels support industrial structure upgrading. Furthermore, the development of the digital economy has given rise to many new industries [40], and the development of new industries is a crucial driving force for industrial structure upgrading. Due to long-term resource development, the industrial structure of resources-based cities is dominated by natural resources development and primary processing, with apparent characteristics of industrial privatization. The new industries brought about by the digital economy undoubtedly inject endogenous power for industrial structure upgrading.

## 3 Theoretical analysis and hypotheses

### 3.1 The direct impact

With the acceleration of global digitization, the digital economy is closely related to the social economy [41]. Many industries use digital technology to drive production and business

activities. The digital economy has become a continuous driving force to promote industrial structural upgrading.

First, the development of the digital economy is conducive to promoting the upgrading of traditional industries. Digitalization is conducive to optimizing the production process and realizing the innovation of production technology [42]. The wide application of digital technology and digital equipment in all aspects of production, operation, and sales in traditional industries can significantly improve the production efficiency and promote the intelligent development of traditional industries. For agriculture, the application of digital technology has improved traditional agricultural production methods, and the application of remote sensing, the Internet of Things, and other new-generation information technology has promoted the automation of the production process; at the same time, the development of rural e-commerce has accelerated the speed of circulation of agricultural products in the market and significantly improved the efficiency of agricultural product circulation. For industry, the digital economy provides information support for industrial production. The production process will become more innovative and intensive through the automatic monitoring and intelligent control of production data. The service industry has become the most active field of digital innovation in China. The integration of the digital economy and the service industry has increased the technical content of services, contributing significantly to the optimization of the service industry.

Second, the digital economy fosters new industries and development models [42]. With the acceleration of the global process, new industries such as the Internet of Things, big data, artificial intelligence, and high-end equipment manufacturing are emerging. Moreover, digital technology breaks the industrial boundaries and promotes industrial extension. The digital economy has accelerated integration with related industries and formed new business forms.

In addition, changes in demand can affect the industrial structure to a large extent. The consumption structure profoundly affects production. The rapid development of the digital economy has changed consumption to a large extent, and online purchases have become a common way of purchasing. With the advancement of big data technology, platforms can accurately analyze consumers' needs and tendencies and push the appropriate goods for consumers. This process is beneficial to create new consumer demand and promoting upgrading the industrial structure. Based on this, this paper puts forward **hypothesis 1**:

The digital economy is conducive to promoting industrial structure upgrading in resource-based cities.

## 3.2 The impact mechanism

Technological innovation is essential in promoting industrial structure upgrading [43]. Innovating new technology can promote transforming and upgrading all links in the industrial chain [44]. Innovation is an essential engine for promoting industrial upgrading [45], and this main conclusion has been confirmed by most studies [46–48]. Some scholars studied the spatial effect of technological innovation and found that technological innovation positively impacts industrial structure upgrading [49]. Other scholars also argued that scientific and technological innovation is vital in promoting the rationalization of industrial structures [50]. Moreover, technological innovation is conducive to stimulating the emergence of new industries and directly promoting the transformation of industrial structure in the direction of rationalization and advancement. At the same time, technological innovation can promote the flow of production resources from high-efficiency production sectors to low-efficiency production sectors and enhance the exchange of knowledge between industries, which in turn promotes the development of industrial structure [44, 51].

The digital economy has profoundly impacted innovation [52]. First, the digital economy provides essential infrastructure and accelerates technological innovation [53]. Knowledge and talent have an essential impact on innovation activities [54]. The digital economy facilitates the pooling of knowledge and dramatically improves technological innovation [55]. In addition, there are uncertainties and risks in the innovation process, and sufficient financial investment is an essential guarantee for innovation to be carried out [56]. The digital economy has created a more relaxed financing environment, alleviating the financing constraints [57], which guarantees the smooth implementation of innovation activities. Research has also confirmed that the increase in research investment is one of the critical paths for the digital economy to improve innovation capacity [58].

Therefore, technological innovation plays a vital role in promoting industrial structure upgrading, and the digital economy significantly improves innovation. **Fig 1** clearly shows the specific mechanisms at work. Based on the above analysis, this paper proposes **hypothesis 2**:

Innovation is the critical mechanism of the digital economy to promote industrial structure upgrading.

## 4 Data and methods

### 4.1 Data and variables

The dependent variable is industrial structure upgrading. Referring to the related research [47], we adopt the calculation formula: $ind_{it} = \sum Y_{int} \times n, n = 1, 2, 3$. Where $Y$ denotes the share of industrial output in GDP; $i$ represents the $i$-th city, and $t$ represents $t$-th year; $n$ represents the type of industry: $n = 1$ represents the primary industry, $n = 2$ represents the secondary industry, and $n = 3$ represents the tertiary industry. The calculated value reflects the evolution of the primary industry to the secondary and tertiary industries. The larger the value, the higher the level of industrial structure upgrading.

The digital economy serves as the primary independent variable in this study. We estimate the digital economy's development level based on Internet growth and digital financial inclusion [59, 60]. The number of Internet users per 100 people, the percentage of workers in the information transmission computer services and software industry, the number of telecommunications services per capita, and the number of mobile phones per 100 people are some indicators of Internet development. We use the *digital financial inclusion index* of Peking University to measure digital financial inclusion [61].

We also select some control variables: government intervention, human capital, urban investment, and trade, which are measured by government fiscal expenditure, the number of full-time teachers per 10,000 people in higher education, the amount of fixed asset investment

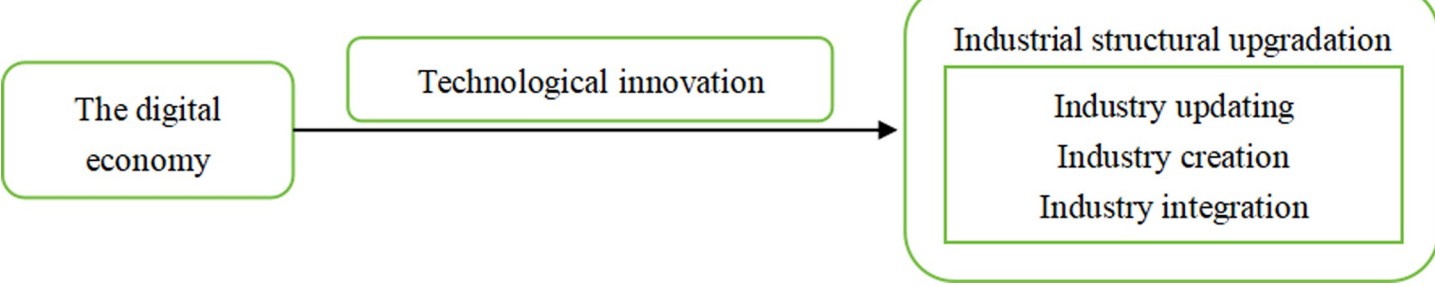

**Fig 1. The effect mechanism of the digital economy on industrial structure upgrading.**

in society, and the import and export trade, respectively. The specific descriptions of the variables are shown in Table 1.

Considering the data availability, this paper uses panel data of 85 resource-based cities in China from 2011 to 2020. All data comes from the "China City Statistical Yearbook", the *Wind* and *EPS* databases. This study mainly uses the interpolation approach or pertinent government reports to fill in individual missing data. We have employed logarithmic processing for all variables.

## 4.2 Methods

In order to assess the impact of digital economy development on the upgrading of industrial structure in resource-based cities, we establish a model as shown below:

$$ind_{it} = \alpha + \beta dig_{it} + \gamma_1 c_1 + \gamma_2 c_2 + \gamma_3 c_3 + \gamma_4 c_4 + \lambda_t + \delta_i + \varepsilon_{it} \tag{1}$$

Where, *indu* stands for the industrial structure upgrading. *dig* represents the level of digital economy development. Four control variables are denoted by $c_1$, $c_2$, $c_3$ and $c_4$, and $\varepsilon$ is a random error term. The subscripts $i$ and $t$ denote city and time, respectively.

To further test the intrinsic mechanism of the digital economy affecting industrial structure upgrading, we also constructed the following mediation effect model:

$$med_{it} = \alpha_0 + \theta_1 dig_{it} + \lambda_1 c_{1it} + \lambda_2 c_{2it} + \lambda_3 c_{3it} + \lambda_4 c_{4it} + \lambda_t + \delta_i + \varepsilon_{it} \tag{2}$$

$$ind_{it} = \alpha_0 + \phi med_{it} + \theta_1 dig_{it} + \lambda_1 c_{1it} + \lambda_2 c_{2it} + \lambda_3 c_{3it} + \lambda_4 c_{4it} + \lambda_t + \delta_i + \varepsilon_{it} \tag{3}$$

# 5 Empirical results

## 5.1 Results of benchmark regression

Table 2 presents the detailed empirical findings. In introducing control variables one by one, the regression coefficient of the digital economy has been significantly positive, indicating that the development of the digital economy promotes the upgrading of the industrial structure of resource-oriented cities.

Due to the excessive dependence on natural resources, the industrial structure of resource-based cities has been relatively homogeneous. Resource extraction, smelting, processing, and other associated sectors are typically their primary industries. Upgrading the industrial structure moves slowly. Global businesses are advancing digital transformation and intelligent manufacturing, and resource-based cities are no exception. The upgrading of the industrial structure must be supported by corresponding financial and technical support. Regarding capital, digital finance broadens financing channels and eases financing constraints, guarantee to

**Table 1. Variable description.**

| Type | Symbol | Variables | Measurement methods |
|---|---|---|---|
| Dependent Variable | $ind$ | industrial structure upgrading | $ind_{it} = \sum Y_{int} \times n, n = 1, 2, 3$ |
| Independent Variable | $dig$ | the digital economy | development level of digital economy |
| Control Variables | $C_1$ | government intervention | government fiscal expenditure |
| | $C_2$ | human capital | the number of full-time teachers per 10,000 people |
| | $C_3$ | urban investment | the amount of fixed asset investment |
| | $C_4$ | trade | import and export trade |

**Table 2. The regression results.**

| Variables | (1) | (2) | (3) | (4) | (5) |
|---|---|---|---|---|---|
| the digital economy | 0.0150** | 0.0144** | 0.0143** | 0.0114* | 0.0122* |
|  | (0.0066) | (0.0066) | (0.0066) | (0.0064) | (0.0063) |
| government intervention |  | 0.0043 | 0.0043 | 0.0019 | 0.0022 |
|  |  | (0.0039) | (0.0039) | (0.0035) | (0.0036) |
| human capital |  |  | 0.0014 | -0.0003 | 0.0007 |
|  |  |  | (0.0045) | (0.0047) | (0.0046) |
| urban investment |  |  |  | 0.0090** | 0.0085** |
|  |  |  |  | (0.0038) | (0.0036) |
| trade |  |  |  |  | 0.0039** |
|  |  |  |  |  | (0.0019) |
| $R^2$ | 0.7391 | 0.7403 | 0.7404 | 0.7487 | 0.7528 |

Note: Robust standard errors are reported in parentheses; ***, **, and * denote significance at 1%, 5%, and 10%, respectively.

upgrade the industrial structure of resource cities. Regarding technology, developing a digital economy can promote the transformation of traditional industries in resource-based cities. The application of digital technology makes the production of traditional industries more efficient, intelligent, and automated, which dramatically improves the quality of products and services in traditional industries; at the same time, the digital economy optimizes the supply chain of traditional industries, which significantly improves the operational efficiency and quality. Moreover, the digital economy facilitates the birth of new industries, and the digital industrial system occupies an increasingly crucial economic position. Therefore, the digital economy shows a facilitating effect on upgrading the industrial structure of resource-based cities.

## 5.2 Robustness tests

To mitigate the possible endogenous problem, we conducted a quasi-natural experiment with the Broadband China strategy. We performed Difference in Difference (DID) estimation, and the results are shown in column (1) of Table 3. In addition, to check the robustness of the benchmark results, we changed the period to 2015–2020. The regression results are shown in column (2) of Table 3. The regression coefficient of the digital economy remains significantly positive, consistent with the benchmark regression results. Therefore, the digital economy significantly promotes to upgrade the industrial structure of resource-based cities.

## 5.3 Heterogeneity analysis

**5.3.1 Heterogeneity of stages of city development.** Referring to the current research [62], we categorize resource-based cities into four groups: rising cities, mature cities, declining cities, and regenerative cities. In order to promote the sustainable development of all types of cities, it is vital to guide them in exploring their development modes and clarifying the development direction and critical tasks of different types of cities. In order to identify the impact of the digital economy on industrial structure upgrading in different types of resource-based cities, we conducted a heterogeneity test. We show the results in Table 4.

Regardless of the type of city, the digital economy has a significant role in promoting industrial structure upgrading. In terms of the extent of its role, the digital economy has a more prominent role in promoting the upgrading of industrial structures in declining cities. Among these four types of cities, declining cities have the lowest level of economic development and

**Table 3. Results of robustness tests.**

| Variables | (1) Did | (2) Changing time interval |
|---|---|---|
| treat*period | 0.0049[*] | —— |
| | (0.0026) | |
| the digital economy | —— | 0.0137[***] |
| | | (0.0042) |
| government intervention | 0.0028 | 0.0039 |
| | (0.0024) | (0.0035) |
| human capital | 0.0010 | 0.0101[***] |
| | (0.0025) | (0.0037) |
| urban investment | 0.0098[***] | 0.0119[***] |
| | (0.0018) | (0.0027) |
| trade | 0.0035[***] | 0.0040[**] |
| | (0.0011) | (0.0017) |

Note: Robust standard errors are reported in parentheses; *, **, and *** represent the 10%, 5%, and 1% significance levels.

face more development challenges. Therefore, from a marginal perspective, the digital economy can have a greater marginal facilitating effect on the industrial structure upgrading of such cities. In mature cities, where resource development is at a stable stage and the level of economy and social development is high, the marginal contribution of the digital economy to the upgrading of the local industrial structure is more limited. The resource development of growing cities is in an upward stage, with strong potential for economic and social development; the economic development mode of regenerative cities has been transformed and stepped into a benign development track. From the perspective of the current state of development, the level of economic development of growing cities and regenerative cities is between that of depleted cities and mature cities, so the marginal contribution of the digital economy to the upgrading of the industrial structure of these two types of cities is at an intermediate level.

**5.3.2 Heterogeneity in geographic distance.** In addition, in the digital economy, cities are becoming more and more interconnected. According to the "trickle-down" theory and "polarization" theory, provincial cities will have an uncertain influence on the development of neighboring cities. Provincial capitals can adversely affect the development of neighboring cities through the siphoning effect but also drive the development of neighboring cities through the spillover effect. Based on this, we categorize the sample into two groups: those close to and those far from the provincial capital city, where resource-based cities are located. The following are the specific division criteria: The average geography distance between all resource-based cities and the capital city was first determined. If the distance between a city and the provincial capital city is longer than this average, the city is defined as far from the provincial capital city. Conversely, the city is classified into the sample group close to the provincial capital

**Table 4. Heterogeneity regression results (different stages of resource development).**

| Variables | (1) Growing city | (2) Mature city | (3) Declining city | (4) Reproducible city |
|---|---|---|---|---|
| the digital economy | 0.0283[**] (0.0122) | 0.0077[*] (0.0095) | 0.0379[***] (0.0117) | 0.0286[*] (0.0140) |
| control variables | Yes | Yes | Yes | Yes |
| $R^2$ | 0.6727 | 0.7998 | 0.5505 | 0.7978 |

Note: Robust standard errors are reported in parentheses; *, **, and *** represent the 10%, 5%, and 1% significance levels.

city. In this study, we chose 85 resource-based cities as samples, of which 35 are far from the provincial capital city, and 53 are close to it. Table 5 displays regression results. The influence of the digital economy will diminish as the distance from the provincial capital city increases. The closer to the provincial capital city, the more pronounced the digital economy's contribution to upgrading the industrial structure. This conclusion can be explained by the borrowing scale between cities. Borrowing scale mainly refers to the fact that small cities can obtain development opportunities by borrowing the scale effect of neighboring big cities. Resource-based cities near provincial capitals can benefit more from the scale effect than those far from provincial capitals. As a result, the digital economy of such cities can realize a higher level of digital economy development, which will have a more significant role in promoting upgrading industrial structures.

## 5.4 Analysis of mediation effect

Technological innovation is closely linked to the evolution of existing industries and the birth of new ones. According to the previous analysis, we use the number of patents granted per 10,000 persons as a proxy for urban innovation. Table 6 displays the regression results.

The innovation and digital economy regression coefficients in column (3) are significantly positive. The coefficient value for the digital economy declines when compared to column (1), revealing that innovation exerts a partial mediating effect.

Innovation and technological progress are crucial in promoting industries' structural upgrading. The digital economy has dramatically facilitated technological innovation. On the one hand, the development of the digital economy eases information asymmetry and greatly facilitates the sharing of innovations.

In addition, the economic development level of resource-oriented cities is low, which hinders innovative activities to a certain extent. The scale effect of the digital economy and digital inclusive finance broaden financing channels, provide financial support for the development of innovative activities, and guarantee the smooth progress of innovative activities. Technological innovation can promote the birth and development of new industrial sectors while enhancing the technological content of traditional industries and ultimately promoting the development of industrial structures in the direction of heightening.

## 6 Conclusions and recommendation

This paper empirically investigates how the digital economy affected upgrading the industrial structure in resource-based cities in China. We also discuss the critical mechanism and heterogeneity. Our main conclusions are: (1) The digital economy significantly contributes to upgrading the industrial structure in resource-based cities, and this conclusion still holds after accounting for possible endogenous issues. (2) For resource-based cities at various phases of extraction of resources, this promotion effect shows significant variation. The digital economy has a more prominent role in promoting upgrading industrial structures in resource-depleted cities, followed

**Table 5. Heterogeneity regression results (the distance from the provincial capital city).**

| Variables | (1) Close to the provincial capital city | (2) Far from the provincial capital city |
|---|---|---|
| the digital economy | 0.0167*** (0.0082) | 0.0062 (0.0105) |
| control variables | Yes | Yes |
| R² | 0.7657 | 0.7602 |

Note: Robust standard errors are reported in parentheses; *, **, and *** represent the 10%, 5%, and 1% significance levels.

**Table 6. Regression results of mediation effect.**

| Variables | (1) | (2) | (3) |
|---|---|---|---|
| the digital economy | 0.0122* (0.0063) | 0.1720** (0.0820) | 0.0073** (0.034) |
| innovation | —— | —— | 0.0134*** (0.0019) |
| control variables | Yes | Yes | Yes |
| $R^2$ | 0.7528 | 0.7346 | 0.2842 |

Note: Robust standard errors are reported in parentheses; *, **, and *** represent the 10%, 5%, and 1% significance levels.

by regenerative and growing cities and, finally, mature cities. Moreover, the closer to the provincial capital city, the more pronounced the promotion of the digital economy. The size of the city acted as a cheerful moderator. (3) Innovation is an effective influence mechanism.

The following recommendations are: (1) Strengthen digital infrastructure construction. The results of the study show that the development of the digital economy is conducive to upgrading industrial structure, and the development of the digital economy cannot be separated from the support of digital infrastructure. Therefore, it is necessary to accelerate the comprehensive construction of digital information infrastructure. The digital economy has a more significant role in promoting industrial structure upgrading for resource-exhausted cities and resource cities far from the central city. Therefore, relevant departments should adequately plan and take reasonable measures to improve digital infrastructure construction to provide a good foundation for upgrading industrial structures. (2) Implement a digital economy development strategy and encourage digitalization. In order to inject new development dynamics into resource-based cities. It is necessary to concentrate on reforming and modernizing established industries and aggressively promoting the deep integration of the digital economy with dominant industries. It is also necessary to adopt some incentive policies to encourage the development of new industries.

## Supporting information

**S1 Data.**
(XLSX)

## Author Contributions

**Conceptualization:** Zhenqiang Li.

**Data curation:** Zhenqiang Li.

**Formal analysis:** Ke Wang.

**Funding acquisition:** Zhenqiang Li.

**Methodology:** Zhenqiang Li, Qiuyang Zhou, Ke Wang.

**Resources:** Zhenqiang Li.

**Software:** Qiuyang Zhou.

**Supervision:** Qiuyang Zhou.

**Writing – original draft:** Qiuyang Zhou.

**Writing – review & editing:** Qiuyang Zhou, Ke Wang.

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
