## [Decision Letter · Decision Letter 0]

15 Aug 2023

PONE-D-23-23633The Impact of the digital economy on industrial transformation and upgrading in resource-based cities: evidence from ChinaPLOS ONE

Dear Dr. Zhou,

Thank you for submitting your manuscript to PLOS ONE. After careful consideration, we feel that it has merit but does not fully meet PLOS ONE’s publication criteria as it currently stands. Therefore, we invite you to submit a revised version of the manuscript that addresses the points raised during the review process.

We look forward to receiving your revised manuscript.

Kind regards,

Liang Zhuang, Ph.D.

Academic Editor

PLOS ONE

Journal Requirements:

“Li Zhenqiang, the National Social Science Foundation of China, 20BGL290”

3.  Thank you for stating the following in your Competing Interests section:  “NO authors have competing interests”

5. We note that Figure 1 in your submission contain map images which may be copyrighted. All PLOS content is published under the Creative Commons Attribution License (CC BY 4.0), which means that the manuscript, images, and Supporting Information files will be freely available online, and any third party is permitted to access, download, copy, distribute, and use these materials in any way, even commercially, with proper attribution. For these reasons, we cannot publish previously copyrighted maps or satellite images created using proprietary data, such as Google software (Google Maps, Street View, and Earth). For more information, see our copyright guidelines: http://journals.plos.org/plosone/s/licenses-and-copyright.

(1) You may seek permission from the original copyright holder of Figure 1 to publish the content specifically under the CC BY 4.0 license.  

6.  Please ensure that you refer to Figure 2 in your text as, if accepted, production will need this reference to link the reader to the figure.

Reviewers' comments:

Reviewer's Responses to Questions

**Comments to the Author**

1. Is the manuscript technically sound, and do the data support the conclusions?

Reviewer #1: Partly

Reviewer #2: Partly

Reviewer #3: Partly

2. Has the statistical analysis been performed appropriately and rigorously? 

Reviewer #1: Yes

Reviewer #2: Yes

Reviewer #3: I Don't Know

3. Have the authors made all data underlying the findings in their manuscript fully available?

Reviewer #1: Yes

Reviewer #2: No

Reviewer #3: Yes

4. Is the manuscript presented in an intelligible fashion and written in standard English?

Reviewer #1: Yes

Reviewer #2: No

Reviewer #3: Yes

5. Review Comments to the Author

Reviewer #1: This paper offers a valuable investigation into the relationship between the digital economy and the industrial structure in resource-based cities. However, several areas could benefit from revision for greater clarity and context. Here are some suggestions for major revisions:

Introduction and Context: The abstract would benefit from a more detailed explanation of the connection between industrial structure and the sustainable development of resource-based cities. Additionally, the role of the digital economy in the transformation of industrial structure could be further elaborated.

Methodology: The methods used in your study could be more thoroughly explained. It is mentioned that panel data from 2011 to 2020 for 85 resource-based cities in China were used. However, further information regarding the selection of these cities, the variables considered, and the specific statistical methods utilized would add to the comprehensibility of the abstract.

Results: Although the findings are presented, a more in-depth analysis of the results, including specific statistics or quantifiable measures, would give the reader a better understanding of the extent to which the digital economy impacts industrial upgrading.

Mechanism of Action: The abstract mentions that innovation is the primary mechanism of action, but it does not elaborate on this. Readers would benefit from an explanation of how and why innovation plays a significant role in this context.

Heterogeneity Analysis: The distinction between cities in various stages of resource development—growth, maturity, decline, and regeneration—is intriguing. More context or examples illustrating these stages and the unique impacts observed at each stage would be beneficial.

Spatial Influence: The conclusion that proximity to the provincial capital city accentuates the impact of the digital economy on industrial structure upgrading is fascinating. Please elucidate on the factors contributing to this observed spatial influence.

In summary, please consider providing more explicit details and explanations throughout your paper. Your topic is of great interest, and further clarifying your methodology, results, and analysis will greatly enhance the value and impact of your research.

Reviewer #2: 1.Some of the literature cited in the literature review section of this article is not representative, some of the authoritative literature in the field is not cited, and the literature review lacks logic.

2.The article lacks the analysis of the impact mechanism of digital economy on the industrial transformation and upgrading in resource-based cities.

3.The authors' use of industrial structure upgrading to measure industrial transformation and upgrading does not capture the meaning of transformation.

4.The digital economy encompasses more than just Internet growth and digital financial inclusion, and there may be limitations to the authors' use of Internet growth and digital financial inclusion to measure the digital economy.

5.DID did not conduct a parallel trend test and did not analyze the impact of the digital economy on the upgrading of the industrial structure of resource-based cities before and after the implementation of the "Broadband China" policy.

6.Regression analysis lacks robustness tests.

7.According to the authors' sample division, which are the resource-based cities close to the provincial capitals? Which are resource-based cities far from the provincial capitals should be clearly stated. However, due to the large number of cities (84), it is recommended that they be marked with different colors in the map.

8.How innovation plays a mediation effect in the analysis of mediation effect should be explained clearly, and what is its influence mechanism? That is, how the digital economy affects the industrial transformation and upgrading of resource-based cities through innovation.

9.The policy recommendations at the end of the article should be more focused, especially on what should be done for different types of resource-based cities and for resource-based cities near and far from provincial capitals.

10.The English language is not expressed accurately and some sentences are grammatically incorrect, requiring careful language revision.

Reviewer #3: Please see the attachment. Please see the attachment. Please see the attachment. Please see the attachment. Please see the attachment. Please see the attachment. Please see the attachment. Please see the attachment.

6. PLOS authors have the option to publish the peer review history of their article (what does this mean?). If published, this will include your full peer review and any attached files.

Reviewer #1: No

Reviewer #2: No

Reviewer #3: **Yes: **Guoliang Fan

---

## [Author Response · Author response to Decision Letter 0]

18 Oct 2023

Reviewer #1: This paper offers a valuable investigation into the relationship between the digital economy and the industrial structure in resource-based cities. However, several areas could benefit from revision for greater clarity and context. Here are some suggestions for major revisions.

Authors’ Response:

The authors gratefully acknowledge the Associate Editor and the Anonymous Reviewers for their detailed and highly constructive criticisms, which greatly helped us to improve the quality and presentation of our manuscript. In the following, we provide detailed, item-by-item, point-by- point responses to all the very interesting issues raised by the Anonymous Reviewers. We have highlighted the main modiﬁcations introduced in the revised manuscript in blue color to help the Associate Editor and the Anonymous Reviewers in ﬁnding the changes made with regards to the previous version. We are indebted to them for their careful assessment and outstanding suggestions for improving our manuscript, which have been extremely helpful in order to enhance its presentation and technical quality.

1. Introduction and Context: The abstract would benefit from a more detailed explanation of the connection between industrial structure and the sustainable development of resource-based cities. Additionally, the role of the digital economy in the transformation of industrial structure could be further elaborated.

Authors’ Response:

Many thanks for your valuable comments. Following your suggestions, we have revised the relevant content. In the introduction, we explain the relationship between industrial structure upgrading and sustainable development of resources-based cities. Regarding the role of the digital economy in industrial structure upgrading, we have analyzed it in detail in the theoretical analysis, including direct and indirect impacts. The details are as follows:

“

1.Introduction

……

……

Based on the development experience of some countries, for resource-based cities to transform, the industrial structure is crucial. It is undeniable that natural resources have played a driving role in the economic growth of resource-based cities. However, the resource curse effect has become increasingly apparent in recent years; while natural resources constitute the comparative advantages of regions, they also create imbalances in regional industrial structure. In resource-based cities, long-term resource development has formed an industrial structure dominated by natural resource development and primary processing. As a result, the industrial structure of resource-based cities is relatively homogeneous and elementary. This industrial structure dominated by resource industries formed under the advantage of resource endowment increases the difficulty for cities to realize sustainable development. Therefore, industrial structure upgrading is of great significance in the sustainable development process of resource cities. Specifically, in terms of the environment, mining, metallurgy and coal processing industries have brought severe environmental pollution problems. In optimizing the industrial structure, new environmental protection technologies and concepts will be applied to all aspects of production, which will undoubtedly help improve the overall ecological environment. In terms of economy, in the process of industrial structure upgrading, the rapid emergence of new high-tech and service industries can inject new vitality into the economic development of resource-based cities. In terms of employment, upgrading the industrial structure can bring more employment opportunities; Especially the development of new industries, which can absorb more labour and promote employment. In addition, in the context of resource scarcity, industrial structure upgrading is also conducive to enhancing the utilization efficiency of resources and promoting the optimal allocation of resources. Overall, industrial structure upgrading is closely related to the long-term development of resource cities. It is necessary to explore the practical path of upgrading the industrial structure of resource cities.

3 Theoretical Analysis and hypotheses

3.1The direct impact

……

……

3.2 The impact mechanism

……

……

”

2. Methodology: The methods used in your study could be more thoroughly explained. It is mentioned that panel data from 2011 to 2020 for 85 resource-based cities in China were used. However, further information regarding the selection of these cities, the variables considered, and the specific statistical methods utilized would add to the comprehensibility of the abstract.

Authors’ Response:

Many thanks for your valuable comments. First, we added a description of the research methodology. Secondly, we introduced the study objects and all variables. The details are as follows：

“

4 Data and Method

 4.1 Data and Variables

……

……

Table 1 Variable description

Type Symbol Variables Measurement methods

Dependent Variable ind Industrial structure upgrading 

Independent Variable dig Digital economy development level of digital economy

Control Variables gov government intervention government fiscal expenditure

 cap human capital the number of full-time teachers per 10,000 people

 inv urban investment the amount of fixed asset investment

 tra trade import and export trade 

4.2 Methods

In order to assess the impact of digital economy development on the upgrading of industrial structure in resource-based cities, we establish a model as shown below:

（1）

Where, indu stands for the industrial structure upgrading. dig represents the level of digital economy development. Four control variables are denoted by c1, c2, c3 and c4, and ε is a random error term. The subscripts i and t denote city and time, respectively.

To further test the intrinsic mechanism of the digital economy affecting industrial structure upgrading, we also constructed the following mediation effect model:

（2）

（3）

”

3. Results: Although the findings are presented, a more in-depth analysis of the results, including specific statistics or quantifiable measures, would give the reader a better understanding of the extent to which the digital economy impacts industrial upgrading.

Authors’ Response:

Many thanks for your valuable and insightful comments. In the revised manuscript, we added explanations for the results of each study.The details are as follows:

“

5 Empirical Results

5.1 Results of Benchmark regression 

……

……

Due to the excessive dependence on natural resources, the industrial structure of resource-based cities has been relatively homogeneous. Resource extraction, smelting, processing, and other associated sectors are typically their primary industries. Upgrading the industrial structure moves slowly. Global businesses are advancing digital transformation and intelligent manufacturing, and resource-based cities are no exception. The upgrading of the industrial structure must be supported by corresponding financial and technical support. In terms of capital, digital finance broadens financing channels and eases financing constraints, guaranteeing the industrial structure upgrading of resource cities. Regarding technology, developing a digital economy can promote the transformation of traditional industries in resource-based cities. The application of digital technology makes the production of traditional industries more efficient, intelligent and automated, which greatly improves the quality of products and services in traditional industries; at the same time, the digital economy optimizes the supply chain of traditional industries, which significantly improves the operational efficiency and quality. Moreover, the digital economy facilitates the birth of new industries, and the digital industrial system occupies an increasingly important economic position. Therefore, the digital economy shows a facilitating effect on the industrial structure upgrading of resource-based cities.

……

……

5.3 Heterogeneity analysis

5.3.1 Heterogeneity of stages of city development

……

Regardless of the type of city, the digital economy has a significant role in promoting industrial structure upgrading. In terms of the extent of its role, the digital economy has a more prominent role in promoting the upgrading of industrial structures in declining cities. Among these four types of cities, declining cities have the lowest level of economic development and face more development challenges. Therefore, from a marginal perspective, the digital economy can have a greater marginal facilitating effect on the industrial structure upgrading of such cities. In mature cities, where resource development is at a stable stage, and the level of economic and social development is high, the marginal contribution of the digital economy to the upgrading of the local industrial structure is more limited. The resource development of growing cities is in an upward stage, with strong potential for economic and social development; the economic development mode of regenerative cities has been transformed and stepped into a benign development track. From the perspective of the current state of development, the level of economic development of growing cities and regenerative cities is between that of depleted cities and mature cities, so the marginal contribution of the digital economy to the upgrading of the industrial structure of these two types of cities is at an intermediate level.

……

5.3.2 Heterogeneity in geographic distance

In addition, in the digital economy, cities are becoming more and more interconnected. According to the "trickle-down" theory and "polarization" theory, provincial cities will have an uncertain influence on the development of neighbouring cities. Provincial capitals can adversely affect the development of neighbouring cities through the siphoning effect but also drive the development of neighbouring cities through the spillover effect. Based on this, we categorize the sample into two groups: those close to and those far from the provincial capital city, where resource-based cities are located. The following are the specific division criteria: The average geography distance between all resource-based cities and the capital city was first determined. If the distance between a city and the provincial capital city is greater than this average, the city is defined as far from the provincial capital city. Conversely, the city is classified into the sample group close to the provincial capital city. In this study, we chose 85 resource-based cities as samples, of which 35 are far from the provincial capital city and 53 are close to it. Table 4 displays regression results. The influence of the digital economy will diminish as the distance from the provincial capital city increases. The closer to the provincial capital city, the more pronounced the digital economy's contribution to upgrading the industrial structure. This conclusion can be explained by the borrowing scale between cities. Borrowing scale mainly refers to the fact that small cities can obtain development opportunities by borrowing the scale effect of neighbouring big cities. Resource-based cities close to provincial capitals can benefit more from the scale effect than those far from provincial capitals. As a result, the digital economy of such cities can realize a higher level of digital economy development, which will have a more significant role in promoting upgrading industrial structures.

5.4 Analysis of mediation effect

……

Innovation and technological progress are crucial in promoting industries' structural upgrading. The digital economy has dramatically facilitated technological innovation. On the one hand, the development of the digital economy eases information asymmetry and greatly facilitates the sharing of innovations.

In addition, the economic development level of resource-oriented cities is low, which hinders innovative activities to a certain extent. The scale effect of the digital economy and digital inclusive finance broaden financing channels, provide financial support for the development of innovative activities, and guarantee the smooth progress of innovative activities. Technological innovation can promote the birth and development of new industrial sectors while enhancing the technological content of traditional industries and ultimately promoting the development of industrial structures in the direction of heightening.

”

4. Mechanism of Action: The abstract mentions that innovation is the primary mechanism of action, but it does not elaborate on this. Readers would benefit from an explanation of how and why innovation plays a significant role in this context.

Authors’ Response:

Many thanks for your valuable and insightful comments. We detail the mechanistic role of innovation in Theoretical Analysis and hypotheses. The details are shown below:

“

3 Theoretical Analysis and hypotheses

……

3.2 The impact mechanism

Technological innovation is essential in promoting industrial structure upgrading (Wang et al., 2020; Yin et al., 2022). The innovation of new technology can promote the transformation and upgrading of all links in the industrial chain (Zhao et al., 2021; Chen et al., 2022). Innovation is an essential engine for promoting industrial upgrading (Zhang et al., 2022) and this main conclusion has been confirmed by most studies (Wang et al., 2021; Song et al. 2022; Li et al., 2022). Wu & Liu (2022) studied the spatial effect of technological innovation and found that technological innovation positively impacts industrial structure upgrading. Wang et al. (2021) also argued that scientific and technological innovation is vital in promoting the rationalization of industrial structures. Moreover, technological innovation is conducive to stimulating the emergence of new industries and directly promoting the transformation of industrial structure in the direction of rationalization and advancement (Yin et al., 2022). At the same time, technological innovation can promote the flow of production resources from high-efficiency production sectors to low-efficiency production sectors (Zhao & Toh, 2022) and enhance the exchange of knowledge between industries, which in turn promotes the development of industrial structure (Xu et al., 2022).

The digital economy has profoundly impacted innovation (Su et al., 2020; Li et al., 2022; Xu & Li, 2021; Tian et al., 2023). First, the digital economy provides essential infrastructure and accelerates technological innovation (Yu et al., 2023). Knowledge and talent have an essential impact on innovation activities (Huang et al., 2023). The digital economy facilitates the pooling of knowledge and dramatically improves technological innovation (Yang & Jia, 2021). In addition, there are uncertainties and risks in the innovation process, and sufficient financial investment is an essential guarantee for innovation to be carried out (Yao & Yang, 2022 ). The digital economy has created a more relaxed financing environment, alleviating the financing constraints (Li et al., 2023 ), which guarantees the smooth implementation of innovation activities. Research has also confirmed that the increase in research investment is one of the critical paths for the digital economy to improve innovation capacity (Huang et al., 2022 ).

Therefore, there is a vital role for technological innovation in promoting industrial structure upgrading, and the digital economy significantly improves the level of innovation. Based on the above analysis, this paper proposes hypothesis 2:

Innovation is the critical mechanism of the digital economy to promote industrial structure upgrading.

”

5. Heterogeneity Analysis: The distinction between cities in various stages of resource development—growth, maturity, decline, and regeneration—is intriguing. More context or examples illustrating these stages and the unique impacts observed at each stage would be beneficial.

Authors’ Response:

Many thanks for your valuable comments. We provide a detailed description in the heterogeneity analysis. The details are as follows:

“

5.3 Heterogeneity analysis

5.3.1 Heterogeneity of stages of city development

 Referring to the current research (Shao et al., 2023), we categorize resource-based cities into four groups: rising cities, mature cities, declining cities, and regenerative cities. In China, mature cities include Zhangjiakou, Datong, Daqing and Panzhihua; Fushun, Jiaozuo and Wuhai are typical declining cities; and regenerative cities include Tangshan, Baotou, Anshan and Nanyang. In order to promote the sustainable development of all types of cities, it is very necessary to guide all types of cities to explore their development modes and to clarify the development direction and critical tasks of different types of cities. In order to identify the impact of the digital economy on industrial structure upgrading in different types of resource-based cities, we conducted a heterogeneity test. We show the results in Table 3. 

Regardless of the type of city, the digital economy has a significant role in promoting industrial structure upgrading. In terms of the extent of its role, the digital economy has a more prominent role in promoting the upgrading of industrial structures in declining cities. Among these four types of cities, declining cities have the lowest level of economic development and face more development challenges. Therefore, from a marginal perspective, the digital economy can have a greater marginal facilitating effect on the industrial structure upgrading of such cities. In mature cities, where resource development is at a stable stage, and the level of economic and social development is high, the marginal contribution of the digital economy to the upgrading of the local industrial structure is more limited. The resource development of growing cities is in an upward stage, with strong potential for economic and social development; the economic development mode of regenerative cities has been transformed and stepped into a benign development track. From the perspective of the current state of development, the level of economic development of growing cities and regenerative cities is between that of depleted cities and mature cities, so the marginal contribution of the digital economy to the upgrading of the industrial structure of these two types of cities is at an intermediate level.

……

……

”

6. Spatial Influence: The conclusion that proximity to the provincial capital city accentuates the impact of the digital economy on industrial structure upgrading is fascinating. Please elucidate on the factors contributing to this observed spatial influence.

Authors’ Response:

Many thanks for your valuable comments. Regarding the explanation of the finding that the closer to the provincial capital city, the greater the contribution of the digital economy to the upgrading of industrial structure, we explain it in terms of the scale of borrowing between cities. The details are shown below:

“

……

5.3.2 Heterogeneity in geographic distance

In addition, in the digital economy, cities are becoming more and more interconnected. According to the "trickle-down" theory and "polarization" theory, provincial cities will have an uncertain influence on the development of neighbouring cities. Provincial capitals can adversely affect the development of neighbouring cities through the siphoning effect but also drive the development of neighbouring cities through the spillover effect. Based on this, we categorize the sample into two groups: those close to and those far from the provincial capital city, where resource-based cities are located. The following are the specific division criteria: The average geography distance between all resource-based cities and the capital city was first determined. If the distance between a city and the provincial capital city is greater than this average, the city is defined as far from the provincial capital city. Conversely, the city is classified into the sample group close to the provincial capital city. In this study, we chose 85 resource-based cities as samples, of which 35 are far from the provincial capital city and 53 are close to it. Table 4 displays regression results. The influence of the digital economy will diminish as the distance from the provincial capital city increases. The closer to the provincial capital city, the more pronounced the digital economy's contribution to upgrading the industrial structure. This conclusion can be explained by the borrowing scale between cities. Borrowing scale mainly refers to the fact that small cities can obtain development opportunities by borrowing the scale effect of neighbouring big cities. Resource-based cities close to provincial capitals can benefit more from the scale effect than those far from provincial capitals. As a result, the digital economy of such cities can realize a higher level of digital economy development, which will have a more significant role in promoting upgrading industrial structures.

……

”

Last but not least, we gratefully thank the Reviewer again for his/her outstanding comments and suggestions, which greatly helped us to improve the technical quality and presentation of our manuscript.

Reviewer #2: 

1.Some of the literature cited in the literature review section of this article is not representative, some of the authoritative literature in the field is not cited, and the literature review lacks logic.

Authors’ Response:

Many thanks for your constructive comments. We have revised the literature review. We have reviewed the existing literature regarding the impact of the digital economy and the factors influencing the upgrading of industrial structure, and we have cited the latest literature. The details are as follows:

“

2 Literature Review

2.1 The impact of the digital economy

……

2.2 Factors affecting the upgrading of industrial structure

……

”

2.The article lacks the analysis of the impact mechanism of digital economy on the industrial transformation and upgrading in resource-based cities.

Authors’ Response:

Many thanks for your valuable comments. We add the analysis of the mechanism of the impact of the digital economy on the upgrading of industrial structure in the theoretical analysis. The specific content is shown below:

“

3 Theoretical Analysis and hypotheses

……

……

3.2 The impact mechanism

Technological innovation is essential in promoting industrial structure upgrading (Wang et al., 2020; Yin et al., 2022). The innovation of new technology can promote the transformation and upgrading of all links in the industrial chain (Zhao et al., 2021; Chen et al., 2022). Innovation is an essential engine for promoting industrial upgrading (Zhang et al., 2022) and this main conclusion has been confirmed by most studies (Wang et al., 2021; Song et al. 2022; Li et al., 2022). Wu & Liu (2022) studied the spatial effect of technological innovation and found that technological innovation positively impacts industrial structure upgrading. Wang et al. (2021) also argued that scientific and technological innovation is vital in promoting the rationalization of industrial structures. Moreover, technological innovation is conducive to stimulating the emergence of new industries and directly promoting the transformation of industrial structure in the direction of rationalization and advancement (Yin et al., 2022). At the same time, technological innovation can promote the flow of production resources from high-efficiency production sectors to low-efficiency production sectors (Zhao & Toh, 2022) and enhance the exchange of knowledge between industries, which in turn promotes the development of industrial structure (Xu et al., 2022).

The digital economy has profoundly impacted innovation (Su et al., 2020; Li et al., 2022; Xu & Li, 2021; Tian et al., 2023). First, the digital economy provides essential infrastructure and accelerates technological innovation (Yu et al., 2023). Knowledge and talent have an essential impact on innovation activities (Huang et al., 2023). The digital economy facilitates the pooling of knowledge and dramatically improves technological innovation (Yang & Jia, 2021). In addition, there are uncertainties and risks in the innovation process, and sufficient financial investment is an essential guarantee for innovation to be carried out (Yao & Yang, 2022 ). The digital economy has created a more relaxed financing environment, alleviating the financing constraints (Li et al., 2023 ), which guarantees the smooth implementation of innovation activities. Research has also confirmed that the increase in research investment is one of the critical paths for the digital economy to improve innovation capacity (Huang et al., 2022 ).

Therefore, there is a vital role for technological innovation in promoting industrial structure upgrading, and the digital economy significantly improves the level of innovation. Based on the above analysis, this paper proposes hypothesis 2:

Innovation is the critical mechanism of the digital economy to promote industrial structure upgrading.

”

3.The authors' use of industrial structure upgrading to measure industrial transformation and upgrading does not capture the meaning of transformation.

Authors’ Response:

Many thanks for your valuable comments. We have looked up the literature on industrial transformation and upgrading and read it carefully. As you said, industrial structure upgrading in this paper does not reflect the significance of transformation. We originally planned to include the variable of industrial structure rationalization, the measurement of which requires data on the number of people employed in each industry. However, the statistics of some resources-based cities lack data on the number of employed people in industries. Therefore, considering data availability, we deleted the concept of transformation and defined the research theme as a study of the impact of the digital economy on upgrading industrial structure in resource cities.

4.The digital economy encompasses more than just Internet growth and digital financial inclusion, and there may be limitations to the authors' use of Internet growth and digital financial inclusion to measure the digital economy.

Authors’ Response:

Many thanks for your valuable comments. Indeed, the digital economy involves the development of many aspects, and there are limitations in measuring only two aspects: Internet development and digital finance. Currently, there are many studies on the digital economy, but there is no unified standard for measuring the level of development of the digital economy. Measuring the development level of the digital economy from both Internet development and digital finance is a widely used method. In order to improve the credibility of the method, we cite relevant references in the text. The details are as follows:

“

……

The digital economy serves as the primary independent variable in this study. We estimate the digital economy’s development level based on Internet growth and digital financial inclusion (Liang & Li, 2023; Zhao et al., 2020; Li et al., 2022 )……

”

5.DID did not conduct a parallel trend test and did not analyze the impact of the digital economy on the upgrading of the industrial structure of resource-based cities before and after the implementation of the "Broadband China" policy.

Authors’ Response:

Many thanks for your valuable comments. After revising the manuscript, DID analysis is one of the robustness tests for this manuscript. In the analysis, we performed a test for parallel trends. However, due to space reasons and financial reasons, we did not list the detailed process in the manuscript, but only the results as in the other robustness test.

6.Regression analysis lacks robustness tests.

Authors’ Response:

Many thanks for your valuable comments. We added robustness tests to the manuscript. The details are as follows:

“

5 Empirical Results

……

……

5.2 Robustness tests

To mitigate the possible endogeneity problem, we conducted a quasi-natural experiment with the Broadband China strategy. We performed Difference in Difference (DID) estimation, and the results are shown in column (1) of Table 3. In addition, to check the robustness of the benchmark results, we changed the period to 2015-2020. The regression results are shown in column (2) of Table 3. The regression coefficient of the digital economy remains significantly positive, consistent with the benchmark regression results. Therefore, the digital economy significantly promotes the industrial structure upgrading of resource-based cities.

Table 3 Results of Robustness tests

Variables (1)Did (2)Changing time interval

Treat*period 0.0049* 

 (0.0026) 

The digital economy 0.0137***

 (0.0042)

Government intervention 0.0028 0.0039

 (0.0024) (0.0035)

Human capital 0.0010 0.0101

 (0.0025) (0.0037)

Urban investment 0.0098*** 0.0119***

 (0.0018) (0.0027)

Trade 0.0035*** 0.0040**

 (0.0011) (0.0017)

Note: Robust standard errors are reported in parentheses; *, **, and *** represent the 10%, 5%, and 1% significance levels.

”

7.According to the authors' sample division, which are the resource-based cities close to the provincial capitals? Which are resource-based cities far from the provincial capitals should be clearly stated. However, due to the large number of cities (84), it is recommended that they be marked with different colors in the map.

Authors’ Response:

Many thanks for your valuable comments. A detailed description of how to define cities close to and far from provincial capitals is provided in the manuscript. Regarding labelling resource cities on maps, as the editors suggested, the journal cannot publish copyrighted maps. Therefore, we dropped the use of maps. The details are as follows:

“

5.3 Heterogeneity analysis

……

5.3.2 Heterogeneity in geographic distance

In addition, in the digital economy, cities are becoming more and more interconnected. According to the "trickle-down" theory and "polarization" theory, provincial cities will have an uncertain influence on the development of neighbouring cities. Provincial capitals can adversely affect the development of neighbouring cities through the siphoning effect but also drive the development of neighbouring cities through the spillover effect. Based on this, we categorize the sample into two groups: those close to and those far from the provincial capital city, where resource-based cities are located. The following are the specific division criteria: The average geography distance between all resource-based cities and the capital city was first determined. If the distance between a city and the provincial capital city is greater than this average, the city is defined as far from the provincial capital city. Conversely, the city is classified into the sample group close to the provincial capital city. In this study, we chose 85 resource-based cities as samples, of which 35 are far from the provincial capital city and 53 are close to it.

……

……

”

8.How innovation plays a mediation effect in the analysis of mediation effect should be explained clearly, and what is its influence mechanism? That is, how the digital economy affects the industrial transformation and upgrading of resource-based cities through innovation.

Authors’ Response:

Many thanks for your valuable comments. We explain this finding in mediation effect. The specifics are as follows:

“

5.4 Analysis of mediation effect

……

Innovation and technological progress are crucial in promoting industries' structural upgrading. The digital economy has dramatically facilitated technological innovation. On the one hand, the development of the digital economy eases information asymmetry and greatly facilitates the sharing of innovations.

In addition, the economic development level of resource-oriented cities is low, which hinders innovative activities to a certain extent. The scale effect of the digital economy and digital inclusive finance broaden financing channels, provide financial support for the development of innovative activities, and guarantee the smooth progress of innovative activities. Technological innovation can promote the birth and development of new industrial sectors while enhancing the technological content of traditional industries and ultimately promoting the development of industrial structures in the direction of heightening.

”

9.The policy recommendations at the end of the article should be more focused, especially on what should be done for different types of resource-based cities and for resource-based cities near and far from provincial capitals.

Authors’ Response:

Many thanks for your valuable comments. We have revised policy recommendations. The details are set out below:

“

6 Conclusions and Recommendation

This paper empirically investigates how the digital economy affected upgrading the industrial structure in resource-based cities in China. We also discuss the critical mechanism and heterogeneity. Our main conclusions are: (1) Digital economy significantly contributes to upgrading the industrial structure in resource-based cities, and this conclusion still holds after accounting for possible endogenous issues. (2) For resource-based cities at various phases of extraction of resources, this promotion effect shows significant variation. The digital economy has a more prominent role in promoting upgrading industrial structures in resource-depleted cities, followed by regenerative and growing cities and, finally, mature cities. Moreover, the closer to the provincial capital city, the more pronounced the promotion of the digital economy. The size of the city acted as a cheerful moderator. (3) Innovation is an effective influence mechanism.

The following recommendations are: (1) Strengthen digital infrastructure construction. The results of the study show that the development of the digital economy is conducive to upgrading industrial structure, and the development of the digital economy cannot be separated from the support of digital infrastructure. Therefore, it is necessary to accelerate the comprehensive construction of digital information infrastructure. For resource-exhausted cities and resource cities far away from the central city, the digital economy has a greater role in promoting industrial structure upgrading. Therefore, relevant departments should adequately plan and take reasonable measures to improve digital infrastructure construction to provide a good foundation for upgrading industrial structures. (2) Implement a digital economy development strategy and encourage digitalization. In order to inject new development dynamics into resource-based cities. It is necessary to concentrate on reforming and modernizing established industries and aggressively promoting the deep integration of the digital economy with dominant industries. It is also necessary to adopt some incentive policies to encourage the development of new industries.

”

\\10.The English language is not expressed accurately and some sentences are grammatically incorrect, requiring careful language revision.

Authors’ Response:

Many thanks for your valuable comments. After completing all revisions, we submitted the manuscript to a professional language agency for linguistic touch-ups.

Last but not least, we gratefully thank the Reviewer again for his/her outstanding comments and suggestions, which greatly helped us to improve the technical quality and presentation of our manuscript.

Reviewer #3: 

The paper presents a research on the influence of the digital economy to the upgrading industrial structure on resource-based cities in China. By using panel data from 2011 to 2020 for 85 resource-based cities in China, the research found that the digital economy is benefit to the upgrading of industrial structure and innovation is the primary mechanism of action. According to the heterogeneity analysis, the results show that there are differences in the impact of the digital economy at the different stages of resource development and the different distance between provincial capital city to resource cities. The paper is well written and the research is well conducted，but there are still some problems that could be improved.

Authors’ Response:

We thank the reviewer for his/her outstanding comments and suggestions, which greatly helped us to improve the technical quality and presentation of our manuscript. We have revised the manuscript accordingly, and the detailed corrections are listed below in point-by-point fashion.

Point 1: It is better to add some past literature reviews on the impact of the digital economy on industrial structure upgrading to make the context more coherent.

Authors’ Response:

Many thanks for your valuable comments. In the theoretical analysis, combined with the existing literature, we analyze the specific impact of digital economy on industrial structure upgrading in detail. The specific content is shown as follows:

“

2 Literature Review

2.1 The impact of the digital economy

……

……

With the development of digitalization, the impact of the digital economy on industrial structure upgrading has also attracted attention. Wu and Shao (2022) found that the digital economy promotes industrial structure upgrading. Guan et al. (2021) further found that the impact of the digital economy on upgrading industrial structure has a nonlinear characteristic. Technological innovation is a critical internal mechanism in the digital economy affecting industrial structure upgrading (Zhao et al., 2021; Su et al., 2020).

……

”

“

3 Theoretical Analysis and hypotheses

3.1The direct impact

With the acceleration of global digitization, the digital economy is closely related to the social economy (Ghasemaghaei & Calic, 2019; Li et al., 2022). Many industries utilize digital technology to drive production and various business activities (Zhao et al., 2021). The digital economy has become a continuous driving force to promote industrial structural upgrading (Du et al., 2021).

First, the development of the digital economy is conducive to promoting the upgrading of traditional industries. Digitalization is conducive to optimizing the production process and realizing the innovation of production technology (Miao, 2021). The wide application of digital technology and digital equipment in all aspects of production, operation and sales in traditional industries can significantly improve production efficiency and promote the intelligent development of traditional industries. For agriculture, the application of digital technology has improved traditional agricultural production methods, and the application of remote sensing, the Internet of Things and other new-generation information technology has promoted the automation of the production process; at the same time, the development of rural e-commerce has accelerated the speed of circulation of agricultural products in the market and significantly improved the efficiency of agricultural product circulation. For industry, the digital economy provides information support for industrial production. The production process will become more innovative and more intensive through the automatic monitoring and intelligent control of production data. The service industry has become the most active field of digital innovation in China. The integration of the digital economy and the service industry has increased the technical content of services, contributing significantly to the optimization of the service industry.

Second, the digital economy fosters new industries and development models (Miao, 2021; Ding et al., 2021). With the acceleration of the global digitalization process, new industries such as the Internet of Things, big data, artificial intelligence, and high-end equipment manufacturing are emerging. Moreover, digital technology breaks the industrial boundaries and promotes industrial extension. The digital economy have accelerated their integration with related industries and formed new business forms. 

In addition, changes in demand can affect the industrial structure to a large extent. The consumption structure profoundly affects production. The rapid development of the digital economy has changed the way of consumption to a large extent, and online purchase has become a common way of purchase. With the advancement of big data technology, platforms can accurately analyze consumers' needs and tendencies and push the appropriate goods for consumers. This process is beneficial to creating new consumer demand and promoting the industrial structure upgrading. Based on this, this paper puts forward hypothesis 1:

The digital economy is conducive to promoting industrial structure upgrading in resource-based cities.

”

Point 2: It is better to create a variable definition table, so that readers can have a clearer understanding of the explanatory variable, the dependent variable, and the control variable.

Authors’ Response:

Many thanks for your constructive comments. We have added a detailed description of all the variables in Table 1. The details are as follows:

“

4 Data and Method

 4.1 Data and Variables

……

Table 1 Variable description

Type Symbol Variables Measurement methods

Dependent Variable ind Industrial structure upgrading 

Independent Variable dig Digital economy development level of digital economy

Control Variables gov government intervention government fiscal expenditure

 cap human capital the number of full-time teachers per 10,000 people

 inv urban investment the amount of fixed asset investment

 tra trade import and export trade 

”

Point 3: Authors can add more explanation in the distance from the provincial capital city on Section 4.2 to increasing the readability of the article. 

Authors’ Response:

Many thanks for your constructive comments. We added a description of the distance to the capital city in the Heterogeneity analysis. The details are as follows:

“

5.3 Heterogeneity analysis

……

5.3.2 Heterogeneity in geographic distance

In addition, in the digital economy, cities are becoming more and more interconnected. According to the "trickle-down" theory and "polarization" theory, provincial cities will have an uncertain influence on the development of neighbouring cities. Provincial capitals can adversely affect the development of neighbouring cities through the siphoning effect but also drive the development of neighbouring cities through the spillover effect. Based on this, we categorize the sample into two groups: those close to and those far from the provincial capital city, where resource-based cities are located. The following are the specific division criteria: The average geography distance between all resource-based cities and the capital city was first determined. If the distance between a city and the provincial capital city is greater than this average, the city is defined as far from the provincial capital city. Conversely, the city is classified into the sample group close to the provincial capital city. In this study, we chose 85 resource-based cities as samples, of which 35 are far from the provincial capital city and 53 are close to it. Table 4 displays regression results. The influence of the digital economy will diminish as the distance from the provincial capital city increases. The closer to the provincial capital city, the more pronounced the digital economy's contribution to upgrading the industrial structure.

……

”

Point4: Perhaps a placebo test can be performed after completing DID to make the results more stable.

Authors’ Response:

Many thanks for your valuable comments. After revising the manuscript, DID analysis is one of the robustness tests for this manuscript. Indeed, it is advisable to add a placebo test. During the analysis, we conducted tests for parallel trends. However, due to space and financial reasons, we did not show the detailed analysis process in the manuscript but just listed the results as in the other robustness test. The details are as follows:

“

5.2 Robustness tests

……

Table 3 Results of Robustness tests

Variables (3)Did (4)Changing time interval

Treat*period 0.0049* 

 (0.0026) 

The digital economy 0.0137***

 (0.0042)

Government intervention 0.0028 0.0039

 (0.0024) (0.0035)

Human capital 0.0010 0.0101

 (0.0025) (0.0037)

Urban investment 0.0098*** 0.0119***

 (0.0018) (0.0027)

Trade 0.0035*** 0.0040**

 (0.0011) (0.0017)

Note: Robust standard errors are reported in parentheses; *, **, and *** represent the 10%, 5%, and 1% significance levels.

”

Point5: Pay attention to the format of the table, for example, whether the Table 2,3,4,5 needs to indicate the significance of asterisk (*).

Authors’ Response:

Many thanks for your valuable comments. We have checked the format of all the tables in the manuscript.

Point6: There are various types of resource-based cities, and different resources influence the development of the digital economy and the upgrading of industrial structures in distinct ways.

Authors’ Response:

Many thanks for your valuable comments. We provide a detailed explanation of the heterogeneous impacts of the digital economy on the upgrading of the industrial structure of different types of resource-based cities, and propose specific policy recommendations for each type of city.

Last but not least, we gratefully thank the Reviewer again for his/her outstanding comments and suggestions, which greatly helped us to improve the technical quality and presentation of our manuscript.

---

## [Decision Letter · Decision Letter 1]

26 Dec 2023

PONE-D-23-23633R1The impact of the digital economy on industrial structure upgrading in resource-based cities: evidence from ChinaPLOS ONE

Dear Dr. Zhou,

Thank you for submitting your manuscript to PLOS ONE. After careful consideration, we feel that it has merit but does not fully meet PLOS ONE’s publication criteria as it currently stands. Therefore, we invite you to submit a revised version of the manuscript that addresses the points raised during the review process.

We look forward to receiving your revised manuscript.

Kind regards,

Liang Zhuang, Ph.D.

Academic Editor

PLOS ONE

Journal Requirements:

Reviewers' comments:

Reviewer's Responses to Questions

**Comments to the Author**

1. If the authors have adequately addressed your comments raised in a previous round of review and you feel that this manuscript is now acceptable for publication, you may indicate that here to bypass the “Comments to the Author” section, enter your conflict of interest statement in the “Confidential to Editor” section, and submit your "Accept" recommendation.

Reviewer #1: All comments have been addressed

Reviewer #2: (No Response)

2. Is the manuscript technically sound, and do the data support the conclusions?

Reviewer #1: Partly

Reviewer #2: Partly

3. Has the statistical analysis been performed appropriately and rigorously? 

Reviewer #1: N/A

Reviewer #2: Yes

4. Have the authors made all data underlying the findings in their manuscript fully available?

Reviewer #1: Yes

Reviewer #2: Yes

5. Is the manuscript presented in an intelligible fashion and written in standard English?

Reviewer #1: Yes

Reviewer #2: No

6. Review Comments to the Author

Reviewer #1: I appreciate for the authors' revision. The existing paper has original content and worthy for publication in the journal. I can recommend it for a possible publication.

Reviewer #2: The quality of this paper has been improved after revision. However, the literature review still lacks logic in its writing. The authors did not combine the relationship between the digital economy and industrial structure upgrading in the literature review, and also lacked a systematic literature review on the transformation of resource-based cities. In addition, the mechanism of the impact of digital economy on the industrial structure upgrading of resource-based cities is still not clear enough, and the authors can make diagrams or construct mathematical models to explain this important issue in depth. Last but not least, the English language is not expressed accurately and some sentences are

grammatically incorrect, requiring careful language revision.

7. PLOS authors have the option to publish the peer review history of their article (what does this mean?). If published, this will include your full peer review and any attached files.

Reviewer #1: No

Reviewer #2: No

---

## [Author Response · Author response to Decision Letter 1]

27 Dec 2023

Reviewer #1: I appreciate for the authors' revision. The existing paper has original content and worthy for publication in the journal. I can recommend it for a possible publication.

Authors’ Response:

The authors gratefully acknowledge the Associate Editor and the Anonymous Reviewers for their detailed and highly constructive criticisms, which greatly helped us to improve the quality and presentation of our manuscript. 

Last but not least, we gratefully thank the Reviewer again for his/her outstanding comments and suggestions, which greatly helped us to improve the technical quality and presentation of our manuscript.

Reviewer #2: The quality of this paper has been improved after revision. However, the literature review still lacks logic in its writing. The authors did not combine the relationship between the digital economy and industrial structure upgrading in the literature review, and also lacked a systematic literature review on the transformation of resource-based cities. In addition, the mechanism of the impact of digital economy on the industrial structure upgrading of resource-based cities is still not clear enough, and the authors can make diagrams or construct mathematical models to explain this important issue in depth. Last but not least, the English language is not expressed accurately and some sentences are grammatically incorrect, requiring careful language revision.

Authors’ Response:

The authors gratefully acknowledge the Associate Editor and the Anonymous Reviewers for their detailed and highly constructive criticisms, which greatly helped us to improve the quality and presentation of our manuscript. In the following, we provide detailed, item-by-item, point-by- point responses to all the very interesting issues raised by the Anonymous Reviewers. We have highlighted the main modiﬁcations introduced in the revised manuscript in blue color to help the Associate Editor and the Anonymous Reviewers in ﬁnding the changes made with regards to the previous version. We are indebted to them for their careful assessment and outstanding suggestions for improving our manuscript, which have been extremely helpful in order to enhance its presentation and technical quality.

1. The authors did not combine the relationship between the digital economy and industrial structure upgrading in the literature review, and also lacked a systematic literature review on the transformation of resource-based cities. 

Authors’ Response:

Many thanks for your valuable comments. Following your suggestions, We revised the literature review section. We contextualized the transformation of resource-based cities and the relationship between the digital economy and industrial structure upgrading, respectively. The details are as follows:

“

2 Literature Review

2.1 Transformation of resource-based cities

Many scholars have studied the transformation paths of resource-based cities (Wu et al., 2020; Gu et al., 2022; Sun et al., 2022). Ecological science, innovation, urbanization development, technology, and political incentives all have some influence on the transformation of resource-based cities (Guo et al., 2019; Zhao et al., 2021; Zhang et al., 2018). The government plays a crucial role in the transformational development process of resource-based cities (Liu et al., 2012; Zhang et al., 2018). On the one hand, an appropriate compensation mechanism is essential for resource-based cities (Liu & Zhuang, 2011). Environmental regulation, policy compensation, and financial help are essential for transforming resource-based cities (Lashitew & Werker, 2020; Jiang et al., 2021; Li et al., 2018). In China, government transfer payments promote low-carbon development in resource-based cities (Li & Wang, 2022). On the other hand, laws and regulations can improve the transition performance of resource-based cities by constraining people’s behavior (Li et al., 2013), and the effectiveness of laws and regulations largely depends on the government. In addition, national policies often have a crucial influence on urban transformation (Hu et al., 2016). Several scholars have studied the effects of some policies implemented in China. For example, the civilized city policy has significantly improved energy efficiency in resource-based cities (Li et al., 2022); the resource-based sustainable development policy has contributed to the economic, social, and ecological transformation of resource-based cities (Fan & Zhang, 2021); the new energy demonstration city policy has a significant effect on the green total factor productivity of resource-based cities (Yang et al., 2023). 

2.2 The digital economy and industrial structural upgrading

Industrial structure upgrading implies the process of evolution from primary to advanced level (Yang et al., 2019). With the acceleration of the global digitalization process, the digital economy has become an essential engine for promoting regional industrial structure upgrading (Pang et al., 2022), and resource-based cities are no exception. Due to the excessive reliance on natural resources, the industrial structure of most resource-based cities is relatively single, which crowds out the space for the development of new industries to a large extent. Moreover, the industrial chain of resource-based cities is not long, and there is an extreme lack of high-value-added products, which also restricts the space for upgrading the original industries. The digital economy has injected a new driving force into the industrial structure upgrading of resource-based cities. On the one hand, digital technology is the core driving force of the digital economy (Wu & Shao, 2022). The application of technologies such as big data and artificial intelligence has realized the intelligence of the production process of traditional industries, promoted the digital transformation of traditional industries (Chang et al., 2023), and driven the upgrading of the value chain (Guan et al., 2021). Second, digital finance improves financial services coverage and broadens enterprises' financing channels in resource-based cities (Ren et al., 2023). The broadened coverage and increased financing channels support industrial structure upgrading. Furthermore, the development of the digital economy has given rise to many new industries (Su et al., 2020), and the development of new industries is a crucial driving force for industrial structure upgrading. Due to long-term resource development, the industrial structure of resources-based cities is dominated by natural resource development and primary processing, with apparent characteristics of industrial privatization. The new industries brought about by the digital economy undoubtedly inject endogenous power for industrial structure upgrading.

”

2.In addition, the mechanism of the impact of digital economy on the industrial structure upgrading of resource-based cities is still not clear enough, and the authors can make diagrams or construct mathematical models to explain this important issue in depth.

Authors’ Response:

Many thanks for your valuable comments. Following the reviewers' comments and referring to related studies, we have created diagrams to illustrate the impact mechanisms clearly. The details are as follows：

“

3 Theoretical Analysis and hypotheses

……

……

3.2 The impact mechanism

……

……

”

References: Su, J., Su, K., & Wang, S. (2020). Does the Digital Economy Promote Industrial Structural Upgrading?—A Test of Mediating Effects Based on Heterogeneous Technological Innovation. Sustainability, 13(18), 10105. https://doi.org/10.3390/su131810105

3.Last but not least, the English language is not expressed accurately and some sentences are grammatically incorrect, requiring careful language revision.

Authors’ Response:

Many thanks for your valuable comments. In order to improve the accuracy of English expression, we submitted the manuscript to a professional organization to polish the text.

Last but not least, we gratefully thank the Reviewer again for his/her outstanding comments and suggestions, which greatly helped us to improve the technical quality and presentation of our manuscript.

---

## [Decision Letter · Decision Letter 2]

30 Jan 2024

The impact of the digital economy on industrial structure upgrading in resource-based cities: evidence from China

PONE-D-23-23633R2

Dear Dr. Zhou,

We’re pleased to inform you that your manuscript has been judged scientifically suitable for publication and will be formally accepted for publication once it meets all outstanding technical requirements.

Kind regards,

Liang Zhuang, Ph.D.

Academic Editor

PLOS ONE

Additional Editor Comments (optional):

Reviewers' comments:

Reviewer's Responses to Questions

**Comments to the Author**

1. If the authors have adequately addressed your comments raised in a previous round of review and you feel that this manuscript is now acceptable for publication, you may indicate that here to bypass the “Comments to the Author” section, enter your conflict of interest statement in the “Confidential to Editor” section, and submit your "Accept" recommendation.

Reviewer #1: All comments have been addressed

Reviewer #2: All comments have been addressed

2. Is the manuscript technically sound, and do the data support the conclusions?

Reviewer #1: Partly

Reviewer #2: Yes

3. Has the statistical analysis been performed appropriately and rigorously? 

Reviewer #1: N/A

Reviewer #2: Yes

4. Have the authors made all data underlying the findings in their manuscript fully available?

Reviewer #1: Yes

Reviewer #2: Yes

5. Is the manuscript presented in an intelligible fashion and written in standard English?

Reviewer #1: Yes

Reviewer #2: Yes

6. Review Comments to the Author

Reviewer #1: I appreciate for the authors' revision. The existing paper has original content and worthy for publication in the journal. I can recommend it for a possible publication.

Reviewer #2: The authors have gone through two rounds of revisions, and the quality of the manuscript has been significantly improved. However, the authors' revision of the mechanism of action can still be further optimized. For example, the current mechanism of action diagram is too simple and can be further improved. Overall, the manuscript can be accept.

7. PLOS authors have the option to publish the peer review history of their article (what does this mean?). If published, this will include your full peer review and any attached files.

Reviewer #1: No

Reviewer #2: No

---

## [Editor Report · Acceptance letter]

13 Feb 2024

PONE-D-23-23633R2 

PLOS ONE

Dear Dr. Zhou, 

I'm pleased to inform you that your manuscript has been deemed suitable for publication in PLOS ONE. Congratulations! Your manuscript is now being handed over to our production team.

Kind regards, 

on behalf of

Professor Liang Zhuang

Academic Editor

PLOS ONE